# Anomaly Detection of DC Nut Runner Processes in Engine Assembly

James Simon Flynn [1,*,†] ⓘ, Cinzia Giannetti [1,†] ⓘ and Hessel Van Dijk [2,†]

1    Engineering North, Swansea University Bay Campus, Swansea SA1 8EN, UK
2    Ford Motor Company, Dearborn, MI 48124, USA
*    Correspondence: jflynn63@ford.com
†    These authors contributed equally to this work.

**Abstract:** In many manufacturing systems, anomaly detection is critical to identifying process errors and ensuring product quality. This paper proposes three semi-supervised solutions to detect anomalies in Direct Current (DC) Nut Runner engine assembly processes. The nut runner process is a challenging anomaly detection problem due to the manual nature of the process inducing high variability and ambiguity of the anomalous class. These characteristics lead to a scenario where anomalies are not outliers, and the normal operating conditions are difficult to define. To address these challenges, a Gaussian Mixture Model (GMM) was trained using a semi-supervised approach. Three dimensionality reduction methods were compared in pre-processing: PCA, t-SNE, and UMAP. These approaches are demonstrated to outperform the current approaches used by a major automotive company on two real-world datasets. Furthermore, a novel approach to labelling real-world data is proposed, including the concept of an 'Anomaly No Concern' class, in addition to the traditional labels of 'Anomaly' and 'Normal'. Introducing this new term helped address knowledge gaps between data scientists and domain experts, as well as providing new insights during model development and testing. This represents a major advancement in identifying anomalies in manual production processes that use handheld tools.

**Keywords:** GMM; UMAP; PCA; t-SNE; quality assurance; anomaly detection; nut runner

## 1. Introduction

Developing a system capable of detecting anomalies in production settings is challenging for several reasons. Access to labelled anomaly data is often difficult in production settings where there are often many potential failure modes, each of which is usually rare and difficult to interpret in time-series data [1]. Furthermore, there is a lack of publicly available datasets upon which to develop and test anomaly detection methods in industrial settings.

Manufacturers must therefore develop their own training and testing datasets and solve complex processing and feature engineering challenges that require technical expertise in both data science and the target domain. Not only is this research and development time-consuming, but any given solution may not be transferable to other processes, even if the processes seem similar. These challenges often make it difficult to estimate a Return-On-Investment (ROI) of such data analytics projects. As a result, the value of machine learning solutions is yet to be fully realised in the automotive industry, which typically focuses on short-term ROI projects.

Throughout the engine assembly line, there are various in-process and end-of-line tests to ensure the quality of the final product. At the engine assembly plant where this research was conducted, many of these tests use static process limits to identify potential fault modes. These limits are set by experienced testing engineers with considerable knowledge of the process and are reviewed and updated regularly manually based on recent test data.

In many processes, this visual inspection of process time-series data through a series of dashboards is also an important step in identifying potential process errors. Interviews with engineers on site find that this method has been proven effective in many tests for which the data are clean, well-structured, and highly regular, and the failure modes are well understood. However, process owners recognise that there are opportunities to improve the current anomaly detection processes, for which there are multiple inefficiencies. Firstly, this approach is not well suited to identifying new, previously unseen anomalies where faults may occur within the specified limits. In these cases, current anomaly detection processes are largely reliant on visual inspection, highlighting opportunities to deliver time savings by automating this process. Secondly, current methods require regular tuning whenever the operating parameters of the test or machinery are changed. Automating these processes would reduce the burden on test engineers to evaluate and maintain the current anomaly detection methods. During the interviews, engineers and data scientists also argue that by automating this process using statistical approaches based on historical data, there is an opportunity to significantly increase anomaly detection rates in more complex processes that exhibit high variability.

At Ford Motor Company, a trial is underway to address these opportunities by developing machine learning algorithms to automate or semi-automate anomaly detection across multiple tests in engine assembly. Currently, a single unsupervised algorithm is used to detect anomalies for all processes. This approach uses Principle Component Analysis (PCA) to reduce the dimensionality of the time series data and perform a cluster analysis using Density Based Spacial Clustering (DBSCAN) under the assumption that any noise points are anomalies. This is successful on a range of end-of-line tests, outperforming the current static limit approach. However, the PCA+DBSCAN method is ineffective at identifying anomalies in 'DC Nut Runner' processes.

Nut runner is an assembly process in which anomaly detection is particularly challenging. The process involves a series of nuts being fastened onto the product, either by a manual operator or a machine. An inbuilt torque transducer in the nut runner tool measures torque against time data which can be analysed to detect process anomalies. The process occurs at multiple stations throughout the automotive engine assembly line involving various types of nuts, threads, required torque, process duration, and other process variables. Figure 1 shows an image of a line worker performing a manual nut runner process.

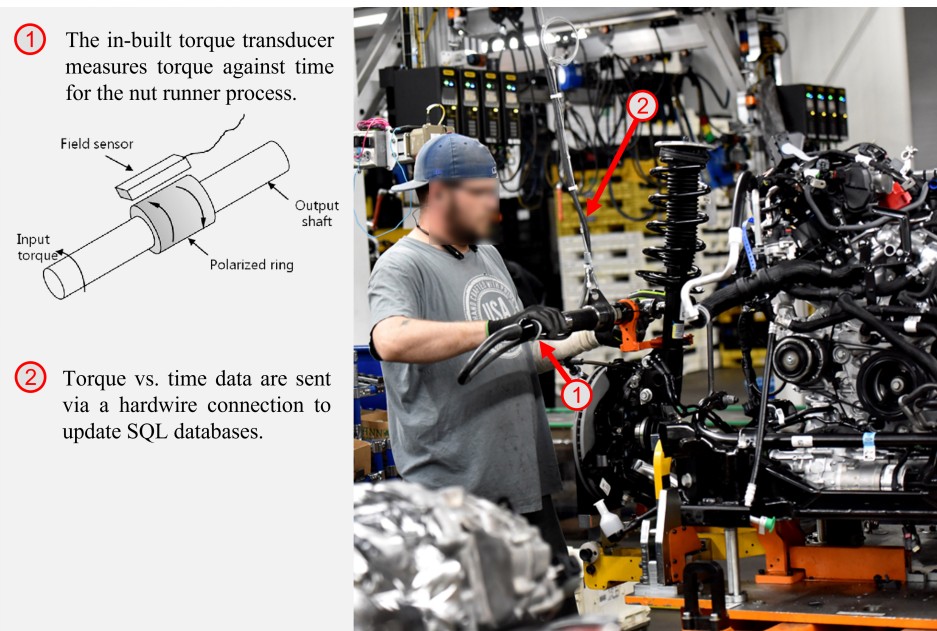

**Figure 1.** An example of a line worker using a DC nut runner tool in engine assembly.

The automated and manual processes produce highly variable data due to the manufacturing variation of the incoming parts and the staged nature of the rundown process. Because the nut runner process involves multiple stages, a human operator may pause for some short duration between stages, resulting in characteristic torque measurements being shifted in time due to these intermittent pauses. Similarly, an automated process may pause between processes for tool changes or geometrical differences between product variants. This staging can be observed in the torque time plots in Figure 2 where torque is applied at different stages of the process, separated by periods of 0 torque, which vary in length. Not only does this staging introduce further variability to the data, but it also removes the cyclicity and seasonality that many methods rely on to identify outliers. This high variability in both the normal data and the anomaly data makes traditional unsupervised clustering approaches such as one-class Support Vector Machines (SVM) and PCA ineffective, as anomalies are not always outliers. Reconstruction methods such as encoder-decoders are also infeasible due to the data being shifted in time at multiple stages, making it difficult to draw a probability distribution from initial data.

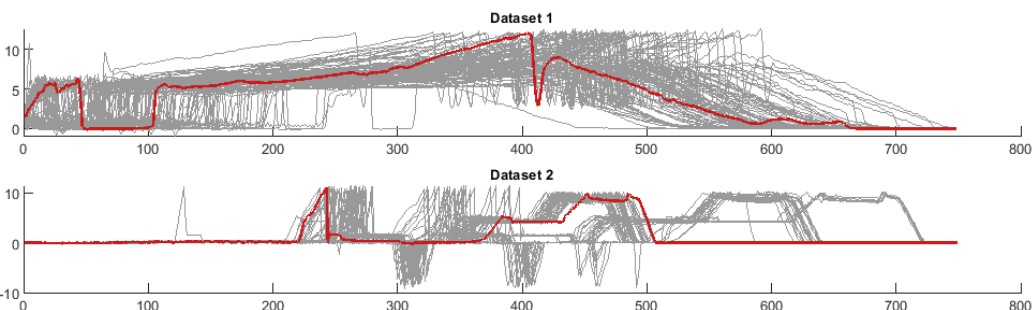

**Figure 2.** Datasets 1 and 2 with a random example of a single observation highlighted in red. Dataset 1 is a manual nut runner process with high variability. Dataset 2 is an automated process where the staging problem can be clearly observed.

This paper presents three semi-supervised clustering approaches to identifying outliers in nut runner data. Each approach uses a Gaussian Mixture Model in combination with different dimensionality reduction methods: PCA, t-SNE, and UMAP. PCA and t-SNE are well-established dimensionality reduction approaches, however, UMAP is a relatively new technique that is emerging as a promising tool but has yet to be explored for application in manufacturing anomaly detection using real-world data. After applying dimensionality reduction, a Gaussian Mixture Model is trained using a semi-supervised approach. The GMM model is a common approach to clustering data that assumes the generative processes to produce the dataset can be described by a mixture of isotropic Gaussian probability density functions. By training the GMM on normal data, threshold regions can be defined, assuming any process that generates anomalies will fall outside of these regions and be identified as an anomaly.

The paper is structured as follows: In Section 2, previous research into time series anomaly detection is reviewed, focusing specifically on semi-supervised approaches and previous applications of GMM and dimensionality reduction approaches. The methodology is presented in Section 3, which includes details on the GMM compared in this study as well as the experimental setup and details on the metrics used to evaluate their performance. The methodology also discusses the challenges of collecting labelled data in real-world manufacturing settings, and how these challenges were overcome by introducing the Anomaly No Concern (ANC) category. The results of the experiments are presented in Section 4 and the discussion is included in Section 5. Finally, the research conclusions are summarised in Section 6.

## 2. Related Research

Anomaly detection is the process of finding, removing, describing, or extracting observations in a dataset that are generated by a different generative process than that of the majority of normal data [2]. Anomaly detection in time-series has been studied by data science researchers for over 50 years in various domains, including fraud detection [3–5], cyber security [6–8], stock market prediction [9], cardiology [2,10–12], engine monitoring [10,11,13], fault detection and condition monitoring [14–17], and manufacturing [18–24]. Approaches to anomaly detection vary greatly on the context of the task, however following several advancements in neural network architectures and computational statistics in the late 1980s and early 1990s, combined with the increased access to the required computational power to apply these methods, the majority of researchers have since focused on some form of machine learning to solve anomaly detection [25–27]. There are three typical approaches for anomaly detection:

- Supervised: Training data are labelled and include both the nominal and anomalous data;
- Clean Semi-Supervised: Training data only include nominal data, while test data also include anomalies;
- Unsupervised: Training data are unlabelled and include both the nominal and anomalous data.

Supervised methods frame the anomaly detection task as a binary classification problem (Normal vs. Anomaly) and use labelled data to train classifiers that distinguish between nominal and anomalous data. This can be effective when the percentage of anomalies $\alpha$ are high ($\alpha > 1\%$). However, in most cases anomalies are very rare ($\alpha < 1\%$), making supervised approaches infeasible as it is both difficult and time-consuming to obtain sufficient labelled anomalous data. Furthermore, the supervised approach makes the assumption that the distribution of anomalous data can be well-defined, and that this distribution can be used to train a statistical model [2]. This assumption is known as the Well-Defined Anomaly Distribution (WDAD) assumption [2]. In manufacturing, this assumption can be utilised to detect repeated machine failures for which the problem space is well understood and sufficient data are available to define the distribution. This is the theoretical basis for Six Sigma practices for which time-invariant data are modelled to fit a well-defined Gaussian distribution and if some measurement exceeds $\pm6\sigma$ from the mean, those instances are flagged as anomalous. Prior research finds that the WDAD assumption is rarely applicable in the real world, as few approaches assume that the anomaly and nominal distribution can be accurately modelled by the analyst [2]. This is especially true in manufacturing environments due to the increasing complexity and variance of data produced by modern manufacturing systems.

In cases where WDAD assumption does not hold, and the fraction of training points that are anomalies are very small ($\alpha < 1\%$), unsupervised or clean semi-supervised methods can be used to detect outliers, although these methods may also fail if anomalies are not outliers or if the distribution of the nominal data has long tails [28].

### 2.1. Types of Anomalies

Anomaly detection of manufacturing systems deals with time-series data and requires different statistical approaches to those used on time-invariant data that assume constant variance and independence of variables. Time series data are a sequence of observations taken by continuous measurement over time, with observations usually collected at equidistant time intervals [29]. Time series data can have properties such as trend, seasonality, cycles and level which can be used to predict future trends and identify anomalies that deviate from the norm.

Much of the existing literature focuses on three types of anomalies in time-series data: point anomalies, collective anomalies, and contextual anomalies [29–31]. Point anomalies are instances where a single point in time deviates significantly from the majority of the time series. An example of a point anomaly in historic weather patterns could be a

single day of heavy snowfall in British springtime. Point anomalies have been studied extensively, with most approaches making the assumption that anomalies are scarce and occur independently of each other [2]. Neural Networks [32], tree-based approaches [33], SVM [8,34], and LSTM [35] have been successfully used to identify point anomalies.

Collective anomalies are where multiple data points in the time series may be considered normal when analysed individually, but when viewed as a collective they demonstrate a pattern of unusual characteristics. Continuing with the weather example, a collective anomaly would be if the snowfall continues for multiple days. Contextual anomalies are cases where data may deviate from the majority of the ANCs but are dismissed as normal due to the context. Contextual anomalies are defined by two attributes [2,3]:

1.  a spacial attribute that describes the local context of a data-point relative to it neighbours;
2.  a behavioural attribute that describes the normality of the data point.

Point et al. provide a detailed mathematical description of contextual anomalies, and how clustering algorithms can be used to identify contextual anomalies in a range of real-world and synthetic data [2]. A common example of contextual anomalies is described using credit card data [3–5]. For example, if an individual's credit card expenditure is significantly high over the course of a week in April it might be considered a collective anomaly and flagged as fraudulent activity. The same transaction behaviour the week before Christmas, however may be considered normal behaviour given the context.

In the example of credit card transactions, we can see that there can often be an overlap between the different types of anomalies. Therefore, it is sometimes necessary to develop a solution that identifies all three types of anomalies. Hundman et al. demonstrate how LSTMs can be used to identify all three types of anomalies in a multivariate time-series dataset to identify spacecraft anomalies in telemetry data [31].

### 2.2. Dimensionality Reduction and Semi-Supervised Clustering

The first step in any anomaly detection task is to use domain knowledge to extract meaningful features from the raw data using feature engineering techniques. These features can then be analysed using a wide range of statistical tools to highlight outliers, which are potential anomalies. The number of meaningful features a dataset has determined whether it has high dimensionality or low dimensionality. As the dimensionality of data increases, it becomes more difficult to draw relationships between these features. This not only requires more training data and more processing power to train models to learn these representations but also makes the trained models more susceptible to overfitting due to noise being present across all dimensions [36].

Dimensionality reduction methods aim to represent high-dimensional data in a lower-dimensional space to visualise data in two or three dimensions and apply cluster analysis approaches that are more suited to lower-dimensional datasets. The most common cluster analysis approaches that have been applied to anomaly detection in time series include: k-means clustering [37–41], Fuzzy C-Means clustering [40,42], Gaussian mixture models [34,37,43], and hierarchical clustering [37,39,41,44].

K-means and Fuzzy C-means clustering involve making initial guesses on the centroid position of a given number of clusters before applying stochastic approaches to iteratively optimise the centroid locations by minimising the distances to points that lie within each centroid's respective clusters. K-means clustering is a hard clustering approach in which each point is assigned to a specific cluster. C-means is a soft clustering approach that assigns individual probabilities to each data point so that data can be assigned to multiple clusters. Diez-Olivan et al. show how k-means clustering can be used for diesel engine condition-based monitoring by detecting anomalies in sensor data [40]. For CBM applications such as this, the normal operating conditions and the anomaly distributions can be well-defined, making cluster analysis a highly effective solution.

A Gaussian mixture model (GMM) is a similar clustering approach that assumes that the process can be described by several sub-processes, each of which may generate a

Gaussian component in the lower dimensional representation [37]. GMM is a probabilistic approach for which maximum likelihood estimation algorithms such as Expectation Maximisation are used for model fitting [37,43].

Unsupervised anomaly detection is a commonly used anomaly detection method. It is often beneficial as it can avoid the need to build high-quality labelled datasets to develop and implement the solution. However, in real-world applications, testing datasets will need to be developed to test and compare models during development to prove their effectiveness before implementation. In cases where the fraction of training points that are anomalies is very small ($\alpha < 1\%$), any testing datasets will be highly imbalanced, with significantly more normal data than anomalies. In these cases, it is practical to utilise this surplus normal data as part of a semi-supervised approach. Previous research has shown that GMMs perform well at semi-supervised anomaly detection in time-series data where the anomaly distribution is not known [45].

Amruthnath et al. compared unsupervised machine learning models to identify anomalies in machine vibration data for predictive maintenance. Of the various clustering methods compared in the study, a combination of PCA and GMM was found to give the best result [37]. However, for this application, the normal operating parameters are well-defined, and only one fault instance was considered. GMM is often applied to analyse biometric time series as it is well suited to handle data with large sample distributions [45,46]. Reddy et al. demonstrate how GMM can be used in unsupervised settings to identify outliers in network traffic data [47]. Reddy et al. applied a semi-supervised approach and discuss the importance of high-quality training data, as the model is sensitive to outliers.

In hierarchical clustering, the initial number of clusters K equals the number of data points. At each iteration, each point is merged with neighbouring clusters until a single cluster is formed. This bottom-up approach is called agglomerative hierarchical clustering and can also be performed in a top-down approach called divisive hierarchical clustering [44]. This process is then used to construct a dendrogram where branches are joined or split at a depth equal to the number of iterations at which those clusters were merged or split. The resulting dendrogram explains the relationship between all the data points in the system and can be horizontally sliced at any point to return the required number of clusters, where small clusters may indicate anomalous system behaviour [37,44].

Dimension reduction techniques can be split into two main categories: Matrix Factorisation and Neighbour Graph approaches. Matrix Factorisation includes algorithms such as Linear Autoencoders, Generalised Low-Rank Models, and Principle Component Analysis (PCA). PCA is one of the oldest and most commonly used methods for dimensionality reduction across a range of scientific disciplines, dating back to work by Pearson in the early 1900s [48]. PCA uses the eigenvectors and eigenvalues of the dataset's covariance matrix to construct linear representations of the data in latent space. These linear representations are called principle components, and those with the highest variance capture the most information of the original data and can be retained for further analysis or plotting while components with low variance can be discarded. PCA has been widely applied in a range of time series anomaly detection tasks by researchers over the past few decades [37,38,40,49,50]. One limitation of PCA is that if the correlations between features are non-linear or unrelated, the resultant transformation may result in false positives or fail to draw any useful relationships [36]. Various tools and add-ins are included in common industrial toolsets, such as Microsoft Excel, that make PCA accessible to engineers.

In recent years, there have been multiple advancements in the development of learning-based neighbour graph algorithms such as t-distributed Stochastic Neighbuor Embedding (t-SNE) [51], and Uniform Manifold Approximation and Projection (UMAP) [52].

t-SNE is a variation of Stochastic Neighbour Embedding first proposed by Hinton and Roweis in 2002 [53]. While PCA retains global structure through eigenvectors with high variance, t-SNE reduces dimensionality by modelling high dimensional data neighbour points as a probability distribution in low dimensional space, thus retaining a more detailed local structure with the loss of some global information. This makes t-SNE favourable

in producing visualisations where understanding this local structure is important and has been used in anomaly detection to visualise bearing faults [54], and superconductor manufacturing errors [55]. Furthermore, t-SNE can reveal non-linear relationships of the data that may be missed using PCA.

UMAP is a recent advancement in dimensionality reduction that has drawn much attention since its publication in 2020, in which Mcinnes et al. propose a topological mapping approach for dimensionality reduction [49,52,56]. UMAP has been shown to improve on t-SNE in preserving both local and global structure of data while also achieving superior run time performance [49,52,56]. UMAP outperformed PCA in clustering time-series data based on cyclical and seasonal characteristics [14,56] and has been used in combination with density-based clustering approaches to highlight periods of anomalous behaviour in time-series data [14,56]. Given the complexity and novelty of UMAP, further research is required to understand the performance of UMAP in industrial settings, with researchers suggesting opportunities for future works in comparing its 2D reduction performance with other distance methods [56].

## 3. Methodology

In this section, the methodology to develop a solution for nut runner anomaly detection is presented. The two datasets were used to develop and test various machine-learning models. The methods used to collect and label these datasets in collaboration with domain experts are discussed, as well as the challenges of these real-world datasets.

Three semi-supervised approaches are presented to detect anomalies in nut runner data. Each method uses a GMM trained on normal data to generate outlier thresholds in a reduced feature space. Three dimensionality reduction approaches were used in combination with the GMM to compare performance. The literature review also finds that LSTMs may be applicable for this case study. However, architectural constraints at the sponsor company presented a barrier to setting up the required Tensorflow GPU environments to explore this solution. This section describes each of these methods, as well as the experimental setup to test and train each of the proposed solutions.

### 3.1. Labeled Data

Nut runner anomalies are rare, and historical process data are not always stored long-term. This makes it challenging to obtain sufficient data on historical machine faults to develop training and testing datasets. If historical fault data do exist, these will still need to be reviewed by a domain specialist to ensure sufficiently high-quality datasets. The task of labelling data is therefore the first major hurdle. Even fully unsupervised methods require high-quality datasets to validate model accuracy. In fault detection applications, training datasets will likely be highly imbalanced as fault instances and anomalies are usually rare. Therefore, large amounts of data may need to be reviewed by domain specialists to gather sufficient data to validate such models.

For this research, a dashboard was developed to speed up the labelling process. The dashboard presented a domain specialist with 12 on-screen examples of time-series nut rundown data to label. Given that anomaly occurrences were presumed to be very rare, the user was informed that all data they were being shown were examples of 'normal' operating conditions. If the user saw any instances that could be considered anomalous, they were asked to label this by using a series of push buttons to categorise the observation into one of three categories:

- True Anomaly: True anomalies are instances where either a known process anomaly has occurred that has compromised part quality, or an unknown anomaly has occurred that requires further inspection before the part is released;
- Anomaly No Concern (ANC): An ANC is defined as an anomalous observation for which no action is needed. This may be because the anomaly can be explained by a known process error that is unlikely to have compromised the quality of the outgoing part;

- Re-hit: A Re-hit is an instance where no data were recorded and the amplitude of the time series remains constant at 0.

If none of the 12 examples on-screen fall within one of these categories, a refresh button is used to label all observations as 'normal', and the display is refreshed with a new batch of 12 images. Figure 3 shows an example of the data labelling dashboard.

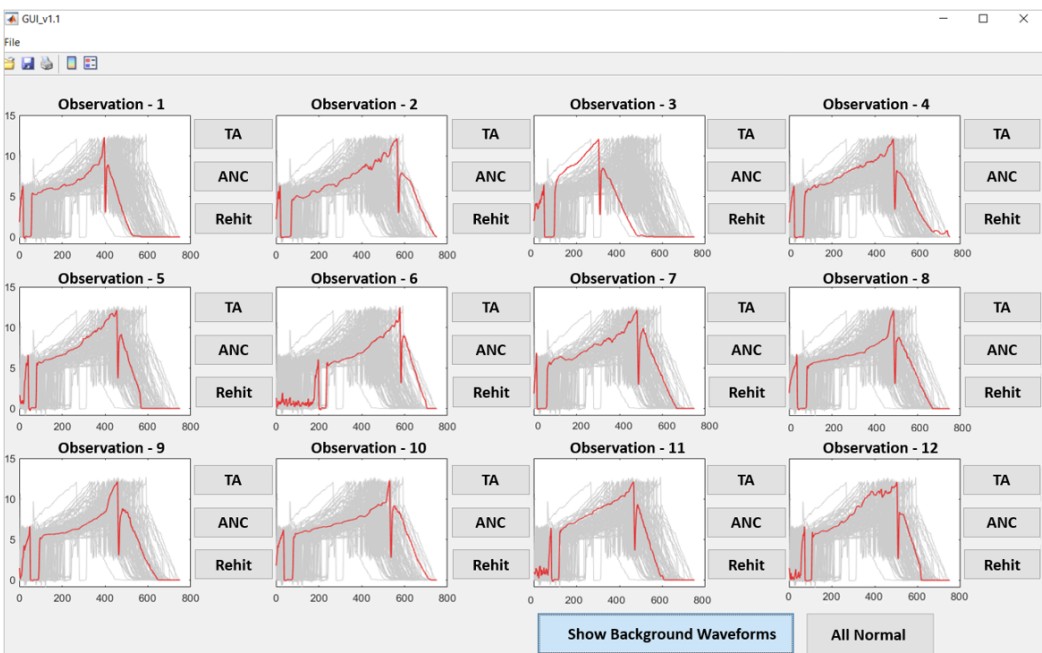

**Figure 3.** The data labelling dashboard allows users to label batches of 12 normal waveforms at a time. For each displayed waveform, users can select True Anomaly (TA), Anomaly No Concern (ANC), or Re-hit. If all waveforms are normal, the All Normal button labels all 12 observations as normal.

This labelling approach proved to be very fast, as less than 1% of processes included True Anomalies and, therefore, most on-screen batches were all normal and labelled 12 at a time. Using this system, domain specialists were able to consistently label data at a rate of approximately 1000 observations per half-hour of labelling. This approach was designed to be used on personal computers to label historical data and was not integrated into any production process or data collection systems.

The ANC class was introduced to overcome some confusion around what constitutes an anomaly. Production line test engineers consider anomalies to be any observation that would result in a part being rejected for inspection/repair. In contrast, the data analysts viewed anomalies as having features or characteristics not found in the majority of the data. To address this contrasting definition of terms, the Anomaly No Concern (ANC) category was included in the labelling process. The judgement between a True Anomaly and an Anomaly No Concern is based on experience, and therefore there is some level of uncertainty and subjectivity in this class. For this reason, the task of labelling is given only to engineers who have a high level of understanding of the process and the test. While this does introduce some uncertainty into our testing datasets, it is notable that engineers will typically err on the side of caution, as product quality is prioritised over all other production metrics. For this reason, it is desirable for any proposed anomaly detection system to flag all instances of both a 'True Anomaly' and an 'Anomaly No Concern'. This being said, a network's ability to detect 'Anomaly No Concern' should never be improved at the expense of reducing the 'True Anomaly' detection rate.

By using this labelling dashboard we were able to overcome two of the major hurdles of developing and implementing machine learning approaches for fault detection: the time taken to label good quality data, and the knowledge gap between domain experts and data

analytic experts. This labelling approach was used to build two testing datasets, one for each process. Three domain experts were given 5000 data to label. When selecting True Anomaly or ANC classes, all data were included where at least one person labelled an observation as an anomaly. When selecting normal data, data were only included when all three agreed on the normal class. Details of each dataset are shown below in Table 1.

**Table 1.** Composition of training and testing datasets to evaluate the performance of the models.

| Dataset | True Anomalies | ANC | Normal | Total |
|---|---|---|---|---|
| Test Dataset 1 | 26 | 37 | 1000 | 1063 |
| Test Dataset 2 | 67 | 100 | 1000 | 1167 |

### 3.2. Machine Learning Model Descriptions

Based on the related research, it was decided to explore the GMM solution for anomaly detection in nut runner data. Three dimension reduction approaches were compared prior to training a GMM: PCA, t-SNE, and UMAP. During the model development phase, the visualisations produced by these approaches proved useful in communicating the results and findings of the nut runner analysis to other team members. This was particularly useful when discussing the importance of high-quality data, and revealed early on that test engineers would often disagree on data labels. Visualising the results through the early development phase made it easier to identify and communicate potential labelling mistakes and to obtain feedback from test engineers. For these reasons, it was decided only to explore 2D representations for all dimensionality reduction approaches.

#### 3.2.1. PCA

Principal Component Analysis (PCA) is a dimensionality reduction technique that aims to preserve the global structure of the data by preserving pairwise distance among all data samples. This is achieved by applying linear mapping using matrix factorization. The mathematical foundations of PCA are widely discussed in previous research and are therefore not discussed in this paper. For further information on the mathematics of PCA, the reader is referred to Syms, 2008 [57].

The current anomaly detection method at Ford Motor Company identifies anomalies using Density-Based Spatial Clustering of Applications with Noise (DBSCAN) to identify noise points in the first two principal components. Figure 4 shows a labelled plot of the first two principal components for datasets 1 and 2. These plots show that, when applying the PCA transform on nut runner data, not all noise points are anomalies, anomalies can form tight clusters, and not all are outliers. In Dataset 1, many anomalies lie within the nominal distribution of the data, sometimes forming small clusters of anomalies within the nominal distribution. In Dataset 2, most anomalies form tight clusters, and the ANC class overlaps significantly with the nominal data.

Because PCA is sensitive to the variance, the data were first normalised. Typically, when applying PCA, the first two or three components are selected, as these components retain the most information on the original structure of the data [58]. However, initial experiments with nut runner data found that using different combinations of principal components resulted in more distinct clusters of anomalies. For this reason, the principle components were also varied when optimising the hyperparameters for the semi-supervised approach.

#### 3.2.2. t-SNE and UMAP

Following the state-of-the-art dimensionality reduction, t-SNE and UMAP were also compared with PCA at clustering the data in 2D before applying the semi-supervised GMM. t-SNE and UMAP are neighbour graph approaches that determine the similarity between the data points before projecting the data onto the lower dimensional space.

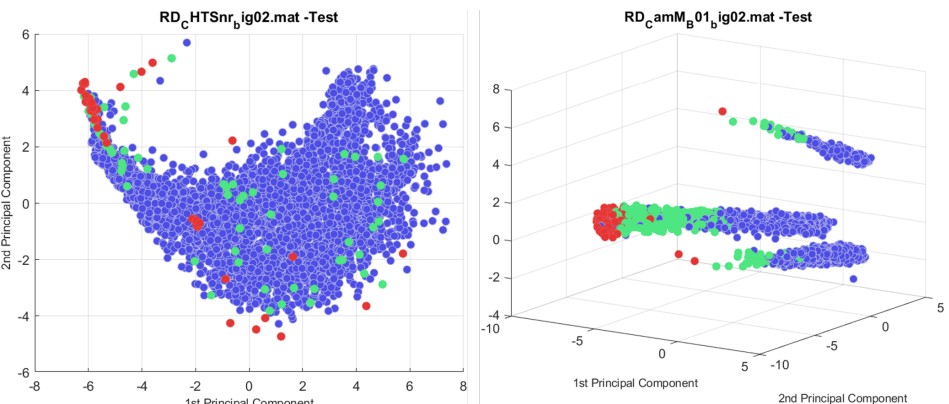

**Figure 4.** A PCA plot of datasets 1 and 2.

Consider some training dataset comprised of $T$ training vectors and $n$ dimensions, given by $X = x_i, \ldots, x_T$, that we wish to map onto a low-dimensional space given by $Y = y_i, \ldots, y_T$. t-SNE is a variation of the SNE algorithm that uses a heavily tailed Student-t distribution in the low-dimensional map, rather than a Gaussian distribution to determine the similarities. For the t-SNE algorithm, conditional probability $q_ij$ is given by:

$$q_{i|j} = \frac{(1 + ||y_i - y_j||^2)^{-1}}{\sum_{k \neq l}(1 + ||y_k - y_l||^2)^{-1}}. \tag{1}$$

By applying a probability distribution with heavy tails, $(1 + ||y_i - y_j||^2)^{-1}$ approaches an inverse square law for large pairwise distances in the low-dimensional representation of the data. This helps retain global structure by separating clusters that are far apart, while retaining local structure within the respective clusters [51]. These new values for $q_{i|j}$ give a gradient of the Kullback–Leibler divergence as:

$$\frac{\delta C}{\delta y_i} = 4 \sum_j (p_{i|j} - q_{i|j})(1 + ||y_i - y_j||^2)^{-1}. \tag{2}$$

The gradient descent is initialised by randomly sampling from an isotropic Gaussian centred on the origin with a small variance. The initial data points $Y$ are then shifted in this low-dimensional space such that the conditional probabilities $Q$ converge on $P$. For further details on how the gradient decent process is optimised to avoid poor local minima, see [51].

Similar to t-SNE, UMAP is also a neighbour graph approach that uses stochastic processes to map $X$ onto $Y$. UMAP is by far the most complex approach discussed in this section, with the theoretical foundations based on manifold theory and topology [52]. At a high level, UMAP applies manifold approximation together with local set representations to map the data onto lower dimensional space. These high-dimensional set representations, known as simplicial sets, describe the high-dimensional feature space by combining multiple simplices defined by the data points $X$. Figure 5 shows a visualisation of these simplices and how they can be combined into a simplicial complex to describe a multi-dimensional feature space. The reader is referred to Mcinnes et al. (2020) for further description of the mathematical description of simplicial complex and how it is used to describe the high-dimensional manifolds [52].

Because t-SNE is a probabilistic approach and both t-SNE and UMAP use stochastic processes, we must combine the training and testing data and perform dimensionality reduction on both datasets to ensure they are mapped onto the same lower dimensional feature space. The 2D outputs of the dimensionality reduction are then split back into the training and testing sets before applying GMM for semi-supervised clustering. Figure 6 shows t-SNE applied to the training and testing nut runner data.

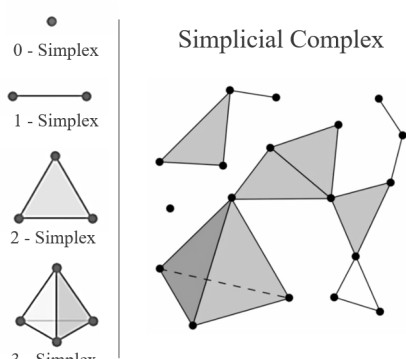

**Figure 5.** UMAP uses combinations of simplicies to provide a simplified representation of the continuous topological space defined by the high dimensional dataset *X* while retaining the global and local structures that define the space.

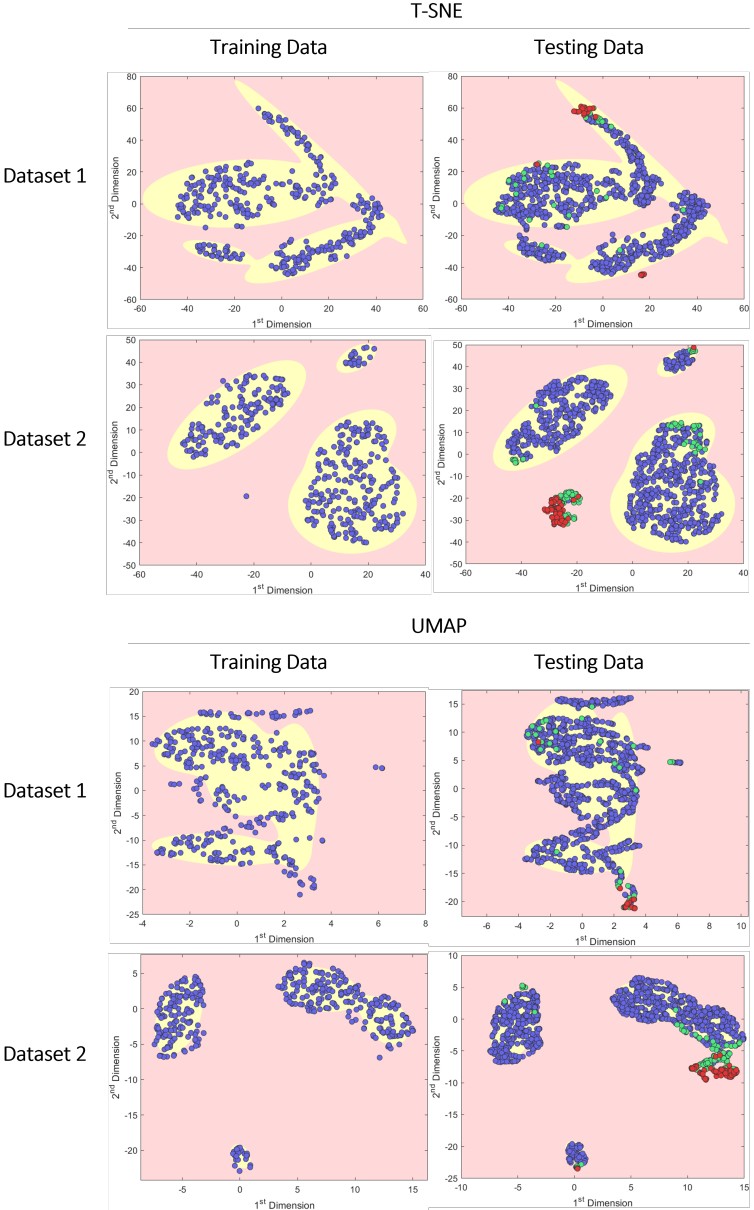

**Figure 6.** The same GMM approach applied using t-SNE and UMAP to reduce and cluster the data. Labelled data include Nominal points (blue), ANC (green), and True Anomalies (red).

For data with very high dimensions, t-SNE has challenges associated with high computational requirements when compared to PCA. The initial construction of the k-NN graph to determine the similarity scores in t-SNE and UMAP is a computational bottleneck for very high dimensional data, and the performance of the k-NN step deteriorates as the dimensionality is increased [59]. Furthermore, the t-SNE method becomes increasingly sensitive to parameter selection as the dimensionality is increased. This also requires users to exhaustively search for optimal parameters which become computationally expensive for very high-dimension datasets [59]. However, in the original paper by [51], the method was shown to have a low error at on a 784-dimensional dataset. This is a higher dimensionality compared to the 750 dimensions in the nut runner time series. Details of the optimization for the t-SNE and UMAP are discussed in Section 3.2.3.

### 3.2.3. GMM

Consider a Gaussian process for which some output *X* is a continuous random variable. It is impossible to define a probability distribution function for all *x*, as there are an uncountably infinite number of potential values. To overcome this, a closely related function can be used to describe the probabilities associated with a continuous random variable [60]. This is called the Probability Density Function (PDF), given by:

$$p(x) = \frac{d}{dx} F(x) \tag{3}$$

$$F(x) = \int_{-\infty}^{x} p(x) dx. \tag{4}$$

A scalar Gaussian component has two parameters that can be used to describe the PDF: the mean $\mu$, and the variance $\sigma^2$. This gives a PDF in the form:

$$p(x|\mu, \sigma^2) = \mathcal{N}(x|\mu, \sigma^2) = \frac{1}{\sqrt{2\pi\sigma^2}} exp\left(\frac{-(x-\mu)^2)}{2\sigma^2}\right). \tag{5}$$

The Gaussian Mixture Model (GMM) assumes that the process can be described by several sub-processes, each of which can be described by a Gaussian probability density with a mean $\mu$, and the variance $\sigma^2$ [37]. However, it is often the case when applying GMM that there are multiple features and high dimensionality [46]. For a multivariate Gaussian with *n* features and *D*-dimensions, a multivariate Gaussian PDF with the same quadratic form is used to describe these components, given by:

$$p(\vec{x}|\vec{\mu}, \Sigma) = \mathcal{N}(\vec{x}|\vec{\mu}_i, \Sigma_i) = \frac{1}{(2\pi)^{D/2}|\Sigma_i|^{1/2}} exp\left(-\frac{1}{2}(\vec{x} - \vec{\mu}_i)'\Sigma_i^{-1}(\vec{x} - \vec{\mu}_i)\right), \tag{6}$$

where $\vec{\mu}$ is the vector mean of length *n*, and $\Sigma$ is the $n \times n$ covariance matrix [46].

The GMM also introduces a scalar weight $w_i$ for each Gaussian component, where $\sum_{i=1}^{M} w_i = 1$. Therefore, a GMM can be described as a weighted sum of *M* Gaussian components, given by:

$$p\vec{x}|\{w_i, \vec{\mu}_i, \Sigma_i\}) = \sum_{i=1}^{M} w_i \mathcal{N}(\vec{x}|\vec{\mu}_i, \Sigma_i), \tag{7}$$

where $i = 1, \ldots, M$. To apply the GMM to make predictions on new data, the model must first be fit to some training dataset comprised of *T* training vectors, given by given by $X = \{\vec{x}_i, \ldots, \vec{x}_T\}$. This is achieved by making initial estimates for the mixture weights $w_i$ mean vectors $\vec{\mu}_i$, and covariance matrices $\Sigma_i$ before optimising these values. The most common approach to optimise the GMM parameters is to use an iterative Expectation–Maximisation (EM) algorithm [46,47,61].

For a $M$ components and with initial estimates for the mixture weights $w_i$ mean vectors $\vec{\mu}_i$, and covariance matrices $\Sigma_i$, the next step in the EM algorithm is to calculate the probability that $\vec{x}_T$ is assigned to component $i$, given by:

$$P_r(i|\vec{x}_t, \gamma\}) = \frac{w_i \mathcal{N}(\vec{x}_t|\vec{\mu}_i, \Sigma_i)}{\sum_{k=1}^{M} w_k \mathcal{N}(\vec{x}_t|\vec{\mu}_k, \Sigma_k)}, \tag{8}$$

where $\gamma = \{w_i, \vec{\mu}_i, \Sigma_i\}$. This probability $P_r(i|\vec{x}_t, \gamma)$ is known as the *A Posterioi* and is used to calculate the next iterations parameters $\gamma'$ using the following equations [46]:

$$w_i' = \frac{1}{T} \sum_{t=1}^{T} P_r(i|\vec{x}_t, \gamma) \tag{9}$$

$$\vec{\mu}_i' = \frac{\sum_{t=1}^{T} P_r(i|\vec{x}_t, \gamma)\vec{x}_t}{\sum_{t=1}^{T} P_r(i|\vec{x}_t, \gamma)} \tag{10}$$

$$\Sigma_i' = \frac{\sum_{t=1}^{T} P_r(i|\vec{x}_t, \gamma)x_t^2}{\sum_{t=1}^{T} P_r(i|\vec{x}_t, \gamma)} - \mu_i'^2. \tag{11}$$

The result of the EM process is dependent on the initialisation points for which to begin the EM optimisation process. This makes the user's selection of the number of Gaussian components important in achieving optimal results. Researchers commonly use methods such as the Bayesian Information Criterion or the Akaike Information Criterion to optimise $M$ [61,62]. Similarly, the result of the GMM is also dependent on the training vectors, which often require careful pre-processing and feature engineering to reduce the dimensionality and cluster the data before applying the GMM. Because of the dependence on the initialisation points, GMM will converge on the local optimum, which may not necessarily be the global optimum. For this reason, multiple runs are required to compare model performance with different. Because the model is deterministic, the best result obtained over multiple runs can then can be used to generate future Gaussian mixture components for classification.

A Gaussian Mixture Model (GMM) was trained in a semi-supervised manner, using 400 normal training data and the number of Gaussian components. For the GMM model, two hyperparameters were optimised using a random search approach—the number of Gaussian components $M$, and the scalar weights $w_i$. The number of Gaussian components $M$ was searched in the range of integers 1 to 6, and the initial estimates for the scalar weights $w_i$ were multiplied by a value in the range of 1 to 3 with intervals of 0.5. When optimising these hyperparameters for the GMM model, hyperparameters were also optimised for the respective dimensionality reduction methods. When applying PCA, the optimum principle components were also included in the random search, considering all possible pairwise combinations of the first ten components. For t-SNE, perplexity was studied in the range of 5 to 50 in steps of 5, and the learn rate in the range of 100 to 1000 with steps of 100; all other values were set as MATLAB defaults. For UMAP, the minimum distance was studied between 0.1 and 0.5 in steps of 0.1, and the number of nearest neighbours was studied between 5 and 25. All other parameters were kept as default in the modified code originally sourced from MATLAB File Exchange [63]. For each combination of hyperparameters, the experiment was repeated three times and the average f-score was calculated. For each model, the random search was stopped after 2 h. This limitation on the optimisation time was decided by those managing the Cloud-based architecture of the anomaly detection pipeline. Because of bandwidth limitations, this 2-h estimate would ensure that, in a worst-case scenario, optimisation runs would still be able to be successfully completed during weekend non-productive time. This would avoid unnecessary downtime of the anomaly detection solution during productive time. At this development stage, all experiments were run locally using a NVIDIA GeForce GTX 1050 GPU.

By training the model using only normal data, the resultant Gaussian approximate components were used to define a threshold boundary to highlight outliers in the testing dataset. Figure 7 shows a plot of datasets 1 and 2 with the outlier thresholds visualised.

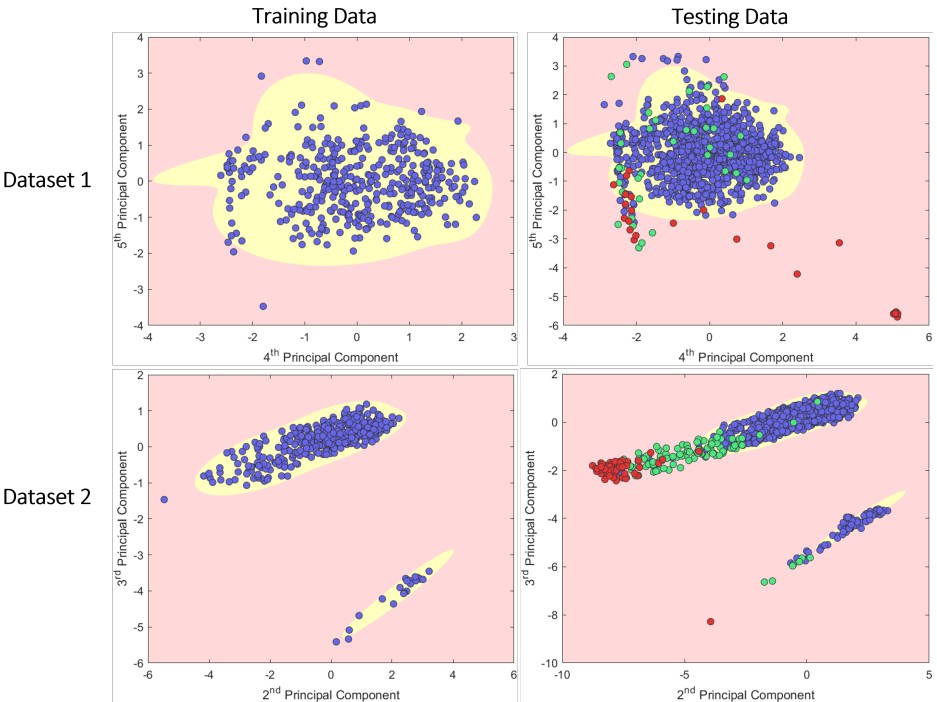

**Figure 7.** Outlier regions calculated using Gaussian mixture model trained on the reduced normal data. Any points that fall in the red area are identified as anomalies. Labelled data includes Nominal points (blue), ANC (green), and True Anomalies (red).

*3.3. Metrics*

In this study, anomalies were treated as the positive class. With this in mind, the following aims are outlined for this study:

- Minimise False Negative Rate: The end goal of this study was to develop an anomaly detection system to improve overall product quality. Therefore, our main objective was to reduce the number of True Anomalies incorrectly identified as Normal;
- Identify a High Percentage of True Anomalies: Reducing the false negative rate should not come at the expense of identifying a low percentage of True Anomalies;
- Near Real Time: Any solution must be able to identify a potential anomalous reading before the part continues onto the next process in the production line. While this time varies between processes, we set a target of under 5 s to perform the analysis;
- Adaptable and Transferable: As processes change over time, any provided solution must be re-trainable with minimal additional development by engineers. Furthermore, any solution must be demonstrated to be effective on multiple nut runner datasets to demonstrate its transferability to multiple use cases.

Given these objectives, we used the F-score false negative rate as our main metric to measure the performance of our methods. The F-score is defined as:

$$F\text{-}score = 2 \times \frac{Precision \times Recall}{Precision + Recall},$$ (12)

where precision is the ratio between true positives and all positives,

$$Precision = \frac{TP}{TP + FP}.$$ (13)

Recall is the measure of the method to correctly identify true positives,

$$Recall = \frac{TP}{TP + FN}. \tag{14}$$

An optimal solution will maximise the F-score while minimising the false positive rate.

These metrics were complicated slightly through the introduction of the ANC class. As discussed in Section 3.1, it can be useful to view ANCs as either an anomaly or a nominal data point depending on the circumstances. For this reason, in any results, we state explicitly whether ANCs are being treated as nominal or as anomalies and discuss the findings within the relevant contexts.

## 4. Results

Typically, when evaluating the performance of models, data scientists at Ford Motor Company treat 'Anomaly No Concern' as 'True Anomalies'. This approach makes sense, as ANCs are still outliers and should be reviewed by test engineers to err on the side of caution and ensure the highest output quality. However, the author argues that this ANC information can provide additional insights into the performance of the models and should be further considered when analysing performance. This is particularly true for nut runner data where the quality of the datasets is difficult to assure given that there is some level of subjective judgement required when labelling the data that affect the overall quality of the datasets. For this reason, two situations are considered:

(a) ANCs are considered as True Anomalies. (Tables 2 and 3);
(b) ANCs are considered to be normal. (Tables 4 and 5).

**Table 2.** Comparison of ML approaches for test Dataset 1a where ANCs are considered as True Anomalies.

| Method | F-Score | Precision | Recall | TP | FN | FP | TN |
|---|---|---|---|---|---|---|---|
| PCA+DBSCAN | 0.14 | 0.14 | 0.14 | 9 | 54 | 55 | 945 |
| PCA+GMM | 0.49 | 0.56 | 0.43 | 27 | 36 | 21 | 979 |
| t-SNE+GMM | 0.25 | 0.35 | 0.19 | 12 | 51 | 22 | 978 |
| UMAP+GMM | 0.27 | 0.17 | 0.68 | 43 | 20 | 216 | 784 |

**Table 3.** Experiment results for Dataset 2a where ANCs are considered as True Anomalies.

| Method | F-Score | Precision | Recall | TP | FN | FP | TN |
|---|---|---|---|---|---|---|---|
| PCA+DBSCAN | 0.23 | 0.49 | 0.15 | 25 | 142 | 26 | 974 |
| PCA+GMM | 0.83 | 0.84 | 0.81 | 136 | 31 | 26 | 974 |
| t-SNE+GMM | 0.73 | 0.73 | 0.74 | 124 | 43 | 47 | 953 |
| UMAP+GMM | 0.59 | 0.48 | 0.77 | 128 | 39 | 137 | 863 |

**Table 4.** Experiment results for Dataset 1 where ANCs are NOT considered as True Anomalies.

| Method | F-Score | Precision | Recall | TP | FN | FP | TN |
|---|---|---|---|---|---|---|---|
| PCA+DBSCAN | 0.09 | 0.06 | 0.15 | 4 | 22 | 60 | 977 |
| PCA+GMM | 0.43 | 0.31 | 0.70 | 18 | 8 | 40 | 1007 |
| t-SNE+GMM | 0.29 | 0.22 | 0.42 | 11 | 15 | 38 | 999 |
| UMAP+GMM | 0.02 | 0.01 | 0.96 | 25 | 1 | 234 | 803 |

**Table 5.** Experiment results for Dataset 2 where ANCs are NOT considered as True Anomalies.

| Method | F-Score | Precision | Recall | TP | FN | FP | TN |
|---|---|---|---|---|---|---|---|
| PCA+DBSCAN | 0.07 | 0.08 | 0.06 | 4 | 63 | 47 | 1053 |
| PCA+GMM | 0.58 | 0.41 | 1.00 | 67 | 0 | 95 | 1005 |
| t-SNE+GMM | 0.61 | 0.45 | 0.97 | 65 | 2 | 78 | 1022 |
| UMAP+GMM | 0.35 | 0.22 | 0.87 | 58 | 9 | 207 | 893 |

The results of the semi-supervised GMM method are largely dependent on the dimensionality reduction approach used to prepare the data. For Dataset 1a, containing manual nut runner data where ANCs are considered True Anomalies, the PCA-GMM performs the best, achieving F-scores of 0.55 compared to 0.25 and 0.27 for t-SNE-GMM and UMAP-GMM respectively. The t-SNE and UMAP approaches are less desirable in comparison, with t-SNE returning a low recall while UMAP returns a low precision. However, all methods outperform the current PCA+DBSCAN used for anomaly detection.

When considering Dataset 1b, where ANCs are considered as normal, the PCA-GMM method sees a drop in true positive rates and false negative rates. The t-SNE approach is the only method for which F-score increases when ANCs are treated as normal data. This aligns with the findings during the model development phases, where t-SNE was found to be particularly useful in identifying true anomalies contaminating the normal training data. However, the method was not as good at distinguishing between ANC and normal data. UMAP-GMM achieves the highest recall of all methods on Dataset 1, however, the very low precision makes the approach undesirable in practice as it would result in considerable added work for test engineers to review these false positives.

For Dataset 2a, PCA-GMM performs the best of all methods, achieving the highest recall and second-highest precision. Furthermore, when considering Dataset 2b, it can be seen that this method identifies 100% of the true anomalies. t-SNE-GMM also performs well on Dataset 2, with similar results to the PCA-GMM method. UMAP-GMM is once again the least best method due to high false positive rates; however, it still achieves a higher F-score than the original PCA-DBSCAN approach.

## 5. Discussion of Results

For both datasets, PCA-GMM performs well and is the most consistent of all methods. This approach performs the best on Dataset 2 identifying 100% of the true anomalies. Unlike t-SNE and UMAP, PCA is deterministic meaning the eigenvectors of the initial transform that gives an optimal result can be easily used to project any new data into the same feature space with minimal computational requirements. Furthermore, the current anomaly detection trial already uses PCA in its current anomaly detection solution, and teams have a good understanding of the work required to further develop and optimize this solution. For these reasons, it was decided to focus continued efforts on the PCA+GMM solutions for nut runner anomaly detection.

Because t-SNE and UMAP are stochastic processes, results will vary between runs, and specific results can be difficult to reproduce. Repeated experiments found that the F-score varied significantly; however, when successful t-SNE produced the most useful visualisations. For example, in Figure 6, the t-SNE results produce a distinct cluster of True Anomalies that were correctly identified and also resulted in a low false positive rate. Furthermore, this run also reveals a mislabelled data point in the training data that appears to be a True Anomaly at $[-20, -20]$. This highlights a major benefit of dimension reduction clustering approaches to produce 2D visualisations. By visualising the data in this abstracted feature space, a quick visual inspection can highlight potential labelling errors in training and testing datasets. These findings are aligned with previous research in which t-SNE was found to be the best method to visualise anomalies in fault detection and manufacturing production data [54,55].

Despite the added value of the visualisations produced by t-SNE and UMAP, the retraining requirements for these algorithms present challenges when considering real-

world implementation due to the variability of results and high computational requirements when compared to PCA. Given that the proposed architecture for the end solution uses Cloud-based services, any additional computational requirements will lead to higher processing costs and may affect the ability of the solution of delivering analysis in near real-time. There are opportunities for future research to explore non-random initialisation options for t-SNE and UMAP that reduce the variability of the final mapping after training. However, this can be complex to implement and may require additional optimisation steps to ensure a solution converges on global minima, rather than local minima. Future research should also explore more efficient Bayesian optimisation approaches that consider more hyperparameters, especially for higher dimensionality datasets greater than 784 dimensions, where the performance of the t-SNE approach is likely to decrease [51].

Although further work is required to apply t-SNE and UMAP for near real-time anomaly detection in nut runner data, our results show that t-SNE and UMAP are still useful tools. Labelling production data is a difficult task, and during our research, it was found that even the most experienced test engineers disagree on True Anomaly and ANC labels, and mistakes are not uncommon when using our labelling tool. Throughout this project, visual inspection of the 2D t-SNE, PCA, and UMAP plots played an important role in cleaning labelled data and highlighting potential labelling errors. The false negative mentioned above in Figure 6 at [−20 −20] has since been confirmed by the test engineer to be an error and is indeed a True Anomaly. Following these results, these methods have since been adopted by Ford Motor Company to validate data labelling efforts as part of this wider project. It is suggested that future research into real-world production data also use t-SNE and UMAP to help clean data before model training if sufficient labelled data are available.

Section 3.1 discusses how part of the reasoning for introducing the ANC category was to overcome some level of subjectivity when labelling data. It is discussed that the training and testing datasets were developed by getting three domain experts to label the data independently. This inevitably results in some disagreement between True Anomaly and ANC classes when labelling data, which is overcome by selecting anomaly data for which at least one person believes it is an anomaly. As more data are compared, this highlights an opportunity for further work to quantify any variability between labelling efforts and how this compares to the resultant accuracy of the models.

It was also mentioned in Section 3 that architectural limitations prevented the use of LSTM in this research. Further work should aim to overcome architectural limitations preventing the implementation of Tensorflow GPU environments.

## 6. Conclusions

This paper proposed three semi-supervised solutions to detect anomalies in nut runner processes. Multiple reasons make nut runner data a challenging anomaly detection problem, including process staging, human-induced variability, and the subjectivity and ambiguity of the anomalous class. These process characteristics lead to an anomaly detection scenario where anomalies are not outliers, and the normal operating conditions are difficult to define. For these reasons, previous unsupervised attempts to automate nut runner anomaly detection have had limited success.

To develop a solution to address these challenges, two bespoke datasets were developed using data collected from two nut runner processes—one manual and one automated. This paper represents the first attempt to identify anomalies in manual manufacturing processes using hand-held tools.

In developing these datasets, a simple user interface and labelling methodology were developed to minimise the human resource requirements to label large amounts of time series data. This dashboard and labelling approach has since been used to support additional projects at Ford Motor Company and is currently being developed as an internal dashboard.

In addition to the data labelling dashboard, a novel concept was introduced to label the training and testing data. When asked to label data, domain experts were given the opportunity to label data as 'Anomaly No Concern', in addition to the traditional labels of 'True Anomaly' and 'Normal'. Introducing this new term helped address knowledge gaps between data scientists and domain experts by highlighting conditions where some processing error had occurred but could be clearly explained as something that would not impact part quality or require any maintenance actions. The inclusion of the ANC class became a key consideration throughout the model development and testing to help clean data, build testing and training data, and address disagreement when labelling data. Furthermore, the ANC class provided further insights into model performance when analysing the results and can be used as further justification for the business case when estimating the solution's impact on quality metrics.

To overcome the challenges of nut runner anomaly detection, multiple solutions were presented that use the available normal data to train machine learning models. These semi-supervised approaches significantly outperform current methods at Ford Motor Company, increasing F-scores by a factor of ten in some cases. The methods presented use a semi-supervised clustering approach, using a combination of dimensionality reduction and GMM for outlier detection. Three dimensionality reduction methods were compared: Principle Component Analysis (PCA), t- Distributed Stochastic Neighbour Embedding (t-SNE), and Uniform Manifold Approximation and Projection (UMAP). Of the three methods, t-SNE and UMAP were found to produce the best visualisations when developing the models, allowing data scientists and domain experts to identify mistakes when labelling data and support data cleaning and model development. However, the combination of PCA and GMM produced the best results when tested on two real-world datasets. To conclude, this paper presents multiple advancements in anomaly detection. This paper represents the first attempt in the academic literature to identify anomalies in manual manufacturing processes that use hand-held tools. A novel concept of an 'Anomaly No Concern' category was introduced to overcome the challenges of labelling real-world data. Multiple semi-supervised clustering approaches were compared, including UMAP, the latest state-of-the-art approach yet to be applied to real-world manufacturing data for anomaly detection. These contributions have led to the successful development of an anomaly detection solution that is currently being implemented at a major automotive company.

**Author Contributions:** J.S.F. conceived of the presented idea with supervision by H.V.D. and C.G. who verified the analytical methods. J.S.F. performed the computations and H.V.D. and C.G. supervised and validated the findings of this work. Writing, editing, and formatting the manuscript was carried out by J.S.F. with support from C.G. Funding acquisition was carried out by C.G. All authors have read and agreed to the published version of the manuscript.

**Funding:** This research was funded by "European Social Fund via the Welsh Government (c80816)" and "the Engineering and Physical Sciences Research Council (Grant Ref: EP/L015099/1)".

**Institutional Review Board Statement:** Not applicable.

**Informed Consent Statement:** Not applicable.

**Data Availability Statement:** Not applicable.

**Acknowledgments:** Cinzia Giannetti acknowledges the support of the UK Engineering and Physical Sciences Research Council (EPSRC) project EP/S001387/1. The authors would also like to acknowledge the M2A funding from the European Social Fund via the Welsh Government (c80816), the Engineering and Physical Sciences Research Council (Grant Ref: EP/L015099/1), and Ford Motor Company that has made this research possible.

**Conflicts of Interest:** The authors declare no conflict of interest.

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
