# Peer review of "Anomaly Detection of DC Nut Runner Processes in Engine Assembly"

_ai, doi:10.3390/ai4010010_

Round 1

Reviewer 1 Report

In this paper, authors presented three semi-supervised approaches to identify outliers in nut runner data that use a Guassian Mixture Model in combination with three dimensionality reduction methods. The present paper is well written and proposed approach is validated with the real data set. However, it looks like authors have not done proper proofread of the paper. Therefore, I suggest proofreading the paper once again. Some of the comments are as follows:

1.      Figure 3 is not very visible.

2.      In line 92 change “The paper is structured follows.: in section 2”. Check this type of typos throughout the paper.

3.      Figure 1 is not cited in the text.

4.      In paragraph 2 of the Introduction section, support some of the claims with appropriate references.

5.      In line 100, Section number is missing.

6.      Authors should remove Appendix A and B, as it looks to be taken from the journal template.

Reviewer 2 Report

The paper studies real world anomaly detection using semi-supervised approaches in comparison to an existing unsupervised approach (PCA+DBSCAN). Overall the paper is well-written and provides the feasibility of applying GMM along with comparing PCA, t-SNE, and UMAP-based dimensionality reduction preprocessing. Authors also provide a labelling mechanism via domain experts since nut runner anomalies are rare. The following are my suggestions to improve the manuscript along with some  minor comments:

1. Figure 1. can be improved by adding another subfigure showing how the time-series data is obtained (a cartoon figure of measurement taken at nut runner setup). Further, Figure 1 seems to be missing a corresponding timeseries example from Dataset 2 , please check and add subfigure.

2. Perhaps add how many domain experts labelled both the datasets and compute cohen's kappa coefficient to understand the variability across experts, say two different domain experts.

3. The AI part is rather standard in the article: GMM is a classical approach. The authors could perhaps add a discussion at the start of 3.2 (before 3.2.1 PCA intro) of other approaches applied in the timeseries world, e.g. LSTM, and why that is not being tried out here (even a vannilla LSTM could have been tried as a baseline).

4. Experimental results look fine and indicate that differentiation of Nominal points, ANC, true anomalies. In Line 443-447 authors mentioned about a cloud based implementation (AWS?). It will be informative to include the spin-off instance details (machine type, GPU allocations etc on the cloud instance). Or the NVIDIA GeForce GTX 1050 GPU is used in the cloud implementation or on the local machine?

5. Authors claim these are novel contributions (Line 597) though GMM, or the best performing PCA-GMM are well known in the timeseries world for many years. It is perhaps novel in applying it to the real world factory (For) setting, though it is not "novel contribution to the anomaly detection literature" would suggest to temper this claim down.

Minor corrections:

1. Expand "DC" at the first instance in introduction.

2. Check Section number at Line 100.

3. Check Ref [63]: Connor Meehan, Jonathan Ebrahimian, Wayne Moore, and Stephen Meehan (2022). Uniform Manifold Approximation and Projection (UMAP) (https://www.mathworks.com/matlabcentral/fileexchange/71902), MATLAB Central File Exchange.

4. Check Abbreviations (Lines 619-622) and add the right ones, and remove Appendix A, B.

Round 2

Reviewer 1 Report

Authors have made significant changes in the manuscript based on my comments. Therefore, I recommend to accept this paper.